# Not All Tokens are Guided Equal: Improving Guidance in Visual Autoregressive Models

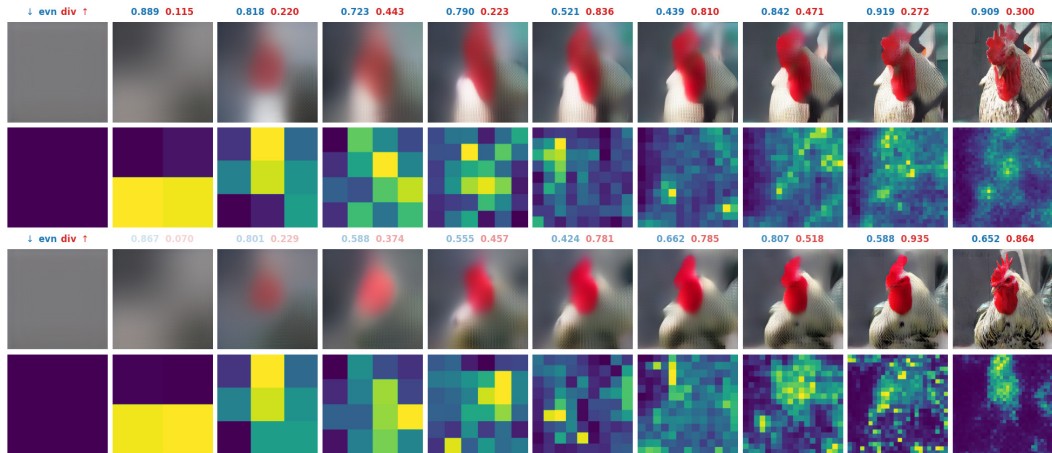

**Figure 1:** Comparison between classifier-free guidance (top) and our method (bottom) on ImageNet $512 \times 512$ class-conditioned generation (class: cock), with VAR (Tian et al., 2024) as the backbone model. Each column corresponds to a sampling step. Each heat map depicts the distribution of guidance on tokens at the respective sampling step, ranging from **purple** (weak guidance) to **yellow** (strong guidance). **Blue** and **red** scores indicate the evenness and divergence at each step, where ↑/↓ indicates that higher/lower is better (see Section 4 for further detail). **Our method improves upon classifier-free guidance by concentrating guidance towards regions of foreground objects.**

## Abstract

Autoregressive (AR) models based on next-scale prediction are rapidly emerging as a powerful tool for image generation, but they face a critical weakness: information inconsistencies between patches across timesteps introduced by progressive resolution scaling. These inconsistencies scatter guidance signals, causing them to drift away from conditioning information and leaving behind ambiguous, unfaithful features. We tackle this challenge with Information-Grounding Guidance (IGG) — a novel mechanism that anchors guidance to semantically important regions through attention. By adaptively reinforcing informative patches during sampling, IGG ensures that guidance and content remain tightly aligned. Across both class-conditioned and text-to-image generation tasks, IGG delivers sharper, more coherent, and semantically grounded images, setting a new benchmark for AR-based methods.

## 1 Introduction

Autoregressive (AR) modelling (Chen et al., 2020; Esser et al., 2021; Ramesh et al., 2021; Li et al., 2024b; Tian et al., 2024; Zhang et al., 2024; Tang et al., 2024; Han et al., 2024; Voronov et al., 2025) has established itself within the field of image generation due to its ability to produce high-quality outputs. AR models, at their core, sample image patches from joint distributions of discrete tokens, allowing generation of complex visual patterns.

A notable advancement in AR modelling is the introduction of multi-scale tokenisation strategies (Tian et al., 2024; Han et al., 2024; Voronov et al., 2025; Zhang et al., 2024; Tang et al., 2024), which enable the models to capture features from coarse to fine levels. This hierarchical approach better aligns with the inherent multi-scale structure of natural images, allowing AR models to effectively model both global structures and fine-grained details. Building on this paradigm, Tian et al. (2024) redefined the autoregressive process as "next-scale" prediction, diverging from "next-token" prediction, which has long been the standard of AR modelling. This methodology—henceforth referred to as *scale-wise* AR (SwAR) modelling as per Voronov et al. (2025); Ren et al. (2024)—is able to generalise features across varying resolutions while effectively reducing the computation of the sampling step, offering a competitive alternative to diffusion-based approaches in terms of sampling quality and time.

Regardless of modelling approaches, the sampling process of generative models relies heavily on various guidance techniques (Kynkäänniemi et al., 2024; Hong et al., 2023; Ho & Salimans, 2022; Dinh et al., 2023a;b; 2024; Karras et al., 2024a). In diffusion-based models, guidance is widely utilised to improve image fidelity at the expense of sample diversity, typically by modulating the denoising trajectory with additional conditioning signals. This mechanism has proven effective in improving visual quality and semantic alignment, particularly in class-conditioned generation tasks. Inspired by the efficacy of diffusion-based guidance, some techniques have been adopted for SwAR modelling. Most notably, Tian et al. (2024) employed classifier-free guidance (Ho & Salimans, 2022) on token predictions (instead of noise and score predictions in diffusion models). Nevertheless, despite impressive empirical results, AR-based guidance is considerably less understood compared to its diffusion-based counterpart. This gap is especially prominent given the fundamental differences between diffusion and AR models.

In light of this, this work is dedicated to investigating the behaviour of AR-based guidance, with a focus on SwAR modelling. In particular, our analysis reveals that guidance in SwAR models is often misaligned, in stark contrast to guidance in diffusion models. From this key insight, we propose IGG (**I**nformation-**G**rounding **G**uidance), a novel guidance scheme designed to accentuate this behaviour, which we posit improves the overall sampling quality of SwAR models. In particular, IGG infers the semantic importance of each token from its surrounding context and applies guidance accordingly. To the best of our knowledge, it is the first method specifically designed for SwAR modelling. We evaluate IGG on various generative tasks. Experimental results consistently surpasses state-of-the-art (SOTA) performances, thereby validating both our hypothesis and the efficacy of our method.

Our contributions in this paper are summarised as follows: **(1)** Present a technical analysis on the dynamics of guidance in diffusion and SwAR modelling, where it was found that guidance tends to be misaligned in the latter. **(2)** Propose IGG, a novel technique that adaptively concentrates guidance on semantically important tokens, and **(3)** Demonstrate the effectiveness of IGG through extensive experiments on class-conditioned generation and text-to-image generation.

## 2 RELATED WORK

**Diffusion models.** Concurrent to the development of AR-based generative models, denoising diffusion models have been receiving much attention for its impressive image generation capability (Rombach et al., 2022; Peebles & Xie, 2023; Karras et al., 2024b). At its core, diffusion-based models learn how to make progressive denoising steps that transform pure noise into the target image. To enhance the quality of the generated samples at the trade-off of diversity, diffusion models rely on guidance methods (see below).

**Autoregressive (AR) models.** AR models have been highly regarded for their success in language tasks. Prior adaptations of these transformer-based architectures for generative tasks Chen et al. (2020); Esser et al. (2021) have been exploring the use of transformers to generate image patches in a raster-scan order. Masked autoregressive models Chang et al. (2022); Li et al. (2024b); Fan et al. (2024) changed this generation behaviour by sampling image patches in a random order at each step. More notably, autoregressive models using the "next-scale prediction" paradigm pioneered by Tian et al. (2024) have been gaining attention for their competitive generation quality to diffusion models.

**Classifier-free guidance (CFG).** Amongst the breadth of guidance methods for generative modelling (Dhariwal & Nichol, 2021; Ho & Salimans, 2022; Hong et al., 2023; Dinh et al., 2023a;b;

2024; Kynkäänniemi et al., 2024; Karras et al., 2024a), CFG (Ho & Salimans, 2022) has remained the de facto technique which has inspired many modern variants in and beyond computer vision, most notably NLP Li et al. (2025); Liao et al. (2025); Zhang et al. (2025). At its core, CFG uses an unconditional model as the distribution to steer the class-conditioned version away from. This mechanism is inspired by classifier guidance (Dhariwal & Nichol, 2021) which opts for an explicit classifier to guide the denoising process. Recent advancements have seen the application of CFG to discrete diffusion and flow models (Nisonoff et al., 2025; Schiff et al., 2025). Given its demonstrated generalisability, CFG has also been adapted to SwAR modelling (Tian et al., 2024) where its efficacy has been empirically verified.

## 3 BACKGROUND

**Scale-wise autoregressive modelling.** Given a vocabulary $\mathcal{V}$, a *token map* $s_k$ is defined as a set of $h_k w_k$ tokens $\{t_{1,1}, t_{1,2}, \cdots, t_{h_k,w_k}\} \subseteq \mathcal{V}$ or, more intuitively, a $h_k \times w_k$ grid of tokens. In its general form, a SwAR model constructs a series of token maps $s := (s_1, \cdots, s_K)$ using an encoder, such that $h_K \times w_K$ matches the resolution of $x$. Then, a model is trained to predict $s$ in an autoregressive manner, maximising the likelihood $p(s|c) = \prod_{k=1}^{K} p(s_k|s_{<k}, c)$, where $c$ denotes the conditioning signal. Finally, a reconstruction of the original $x$ is produced from the predicted token maps using a decoder.

**SwAR-based classifier-free guidance.** The mechanism of classifier-free guidance in SwAR modelling is analogous to diffusion-based modelling, with the only exception in what is being guided (token predictions, in contrast to noise/score predictions). In particular, guidance is applied on each inference step by interpolating between the unconditioned model $p_\theta(\cdot)$ and its conditioned counterpart $p_\theta(\cdot|c)$, which may be expressed as

$$\tilde{p}_\theta(s_k|c) = (1 + \lambda_k)\, p_\theta(s_k|c) - \lambda_k\, p_\theta(s_k), \tag{1}$$

where $w \in \mathbb{R}$ is the guidance scale. Following guidance, sampling can be performed on $p_\theta(s_k|c)$ to obtain the final prediction $\hat{s}_k \in \mathcal{V}^{h_k \times w_k}$. Interestingly, while the original diffusion-based implementation (Ho & Salimans, 2022) fixes the guidance scales $\lambda_1 = \cdots = \lambda_K$, it is possible to generalise it to a guidance *schedule*. For example, Tian et al. (2024) defined the schedule $\lambda_k := w \cdot k/(K-1)$, where $w \in \mathbb{R}$ is a hyperparameter, to gradually accentuate guidance throughout inference. Guidance is typically applied directly on the raw logit outputs of the models in practice. However, throughout the paper, we will occasionally overload the notation $p_\theta(\cdot|c)$ to also denote the associated probabilities, although it should be clear from context which quantities are being referred to.

## 4 ANALYSING THE BEHAVIOUR OF CFG DURING SAMPLING

This section aims to elucidate the key behaviours of CFG and differences between the guidance dynamics of diffusion modelling as opposed to AR modelling, thereby shedding some light on the gap in performance between these two approaches in practice. We begin by offering an alternative perspective to the conventional interpretation of CFG as interpolating between unconditioned and conditioned predictions:

$$\tilde{p}_\theta(s_k|c) = p_\theta(s_k) + \gamma_k\, (p_\theta(s_k|c) - p_\theta(s_k)) =: p_\theta(s_k) + \vec{p}_\theta(s_k|c). \tag{2}$$

Note that Equation 2 becomes equivalent to Equation 1 by setting $\gamma_k := 1 + \lambda_k$. Under this formulation, CFG can be seen as "nudging" the unconditioned predictions towards the conditioned predictions. This interpretation gives a concrete form to the concept of guidance that we can use to validate our hypotheses below. Indeed, it is also consistent with the intention of the authors of the original implementation of CFG, who remarked that the difference between the conditioned and unconditioned score estimates resembles the gradient of an implicit classifier which is guiding the main model (Ho & Salimans, 2022). For ease of notations, we henceforth use $\vec{p}_\theta(\cdot|c)$ to denote the guidance signals characterising these "nudges". A visualisation of these guidance signals is depicted in Figure 1.

**Guidance does not treat tokens equally.** To validate our first hypothesis, we used *Pielou's evenness index* (Pielou, 1966)—or the normalised Shannon entropy—to evaluate the evenness of the guidance

distribution associated with each token map,

$$\text{PEI}(s_k|c) = \frac{H(p_\theta^{\to}(s_k|c))}{\ln(h_k w_k)} \tag{3}$$

Note that $\text{PEI}(\cdot) \in [0, 1]$, where a greater value indicates a more even distribution. In our evaluation, for each image, we compute $\text{PEI}(s_k|c)$ for every sampling step $k$ (although $k = 1$ is omitted for SwAR models because it makes little sense to evaluate a distribution comprising a single value). The overall PEI score for an image is computed as the weighted mean where each sampling step is weighted by its corresponding output resolution (see Figure 2 for a visual illustration). For diffusion models, since the output resolution is constant throughout the sampling process, the weighted mean reduces to the unweighted mean. Comprehensive comparison between a representative model from each paradigm (Table 1) reveals that diffusion modelling exhibits highly uneven guidance signals while SwAR modelling remains lacklustre.

**Guidance prioritises semantically important tokens.** Here, a token is considered to be semantically important if it corresponds to a foreground object or an area within the object with high level of detail in the final generated image. Assuming that this hypothesis indeed holds, it is then natural to assume that the distribution of guidance signals should significantly diverge from the default "unguided" distribution. However, it is not immediately clear how such a distribution would be defined: if we simply disable guidance then the guidance signals would not exist in the first place! Fortunately, it is possible to construct a guidance distribution embodying this hypothetical unguided distribution by sampling guidance signals on tokens that are *not* semantically important. Then, the divergence between the original guided distributions and synthetic unguided distributions may be determined. Indeed, if it does not hold that guidance focusses more on

**Table 1:** Comparison of the evenness (Evn, Equation 3) and divergence (Div, Equation 4) of guidance in EDM2 (Karras et al., 2024b) and VAR (Tian et al., 2024). Results are averaged scores obtained through ImageNet $512 \times 512$ class-conditioned generation. Our method closes the gap between SwAR and diffusion models.

| Model | Evn ($\downarrow$) | Div ($\uparrow$) |
|---|---|---|
| EDM2-S | 0.486 | 0.983 |
| EDM2-XXL | 0.560 | 0.964 |
| VAR-$d36$-CFG | 0.741 | 0.623 |
| VAR-$d36$-IGG | 0.665 | 0.751 |

semantically important tokens, then the nudges applied on background tokens should be roughly equivalent to foreground tokens and the divergence between the two distributions should consequently be low. To validate our second hypothesis, we performed the same comparison experiment. Semantically important tokens were automatically identified by segmenting generated images with YOLOv11 (Khanam & Hussain, 2024) and extracting tokens lying inside the output segmentation masks. Divergence is quantified using *Jensen-Shannon distance* (Endres & Schindelin, 2003),

$$\text{JSD}(p, q) = \sqrt{\tfrac{1}{2}D_{\text{KL}}(p\|m) + \tfrac{1}{2}D_{\text{KL}}(q\|m)}, \tag{4}$$

where $m = \frac{1}{2}(p + q)$ denotes the mixture distribution between $p$ and $q$. Note that $\text{JSD}(\cdot, \cdot)$ also ranges in $[0, 1]$, where a greater value indicates greater divergence. The exact evaluation procedure is outlined in Appendix A.1. Comparison results (Table 1) indicate significant deviation between guided and unguided distributions in the diffusion model, while the SwAR model again struggles to compete.

**Guidance is misaligned in SwAR models.** In addition to the above quantitative evaluations, we also qualitatively compared the guidance dynamics of the two generative modelling paradigms. Visualisation of sampled images from SwAR model revealed numerous instances where guidance signals became progressively dispersed at increasing scale levels, leading to adversarial features in generated images. On the other hand, the diffusion model exhibits guidance signals that are consistently sharp and aligned to semantically important tokens (Figure 2).

## 5 INFORMATION-GROUNDING GUIDANCE

Building on the observations in Section 4, we hypothesise that the potential of CFG has not been fully utilised in SwAR models. In SwAR models, guidance signals often weakens progressively with each sampling step, making it disadvantageous to directly apply guidance strategies deployed in diffusion models, as illustrated in Figure 2. This gap can be narrowed by tuning the guidance signals of AR models so that they more closely align with the evenness and divergence patterns observed

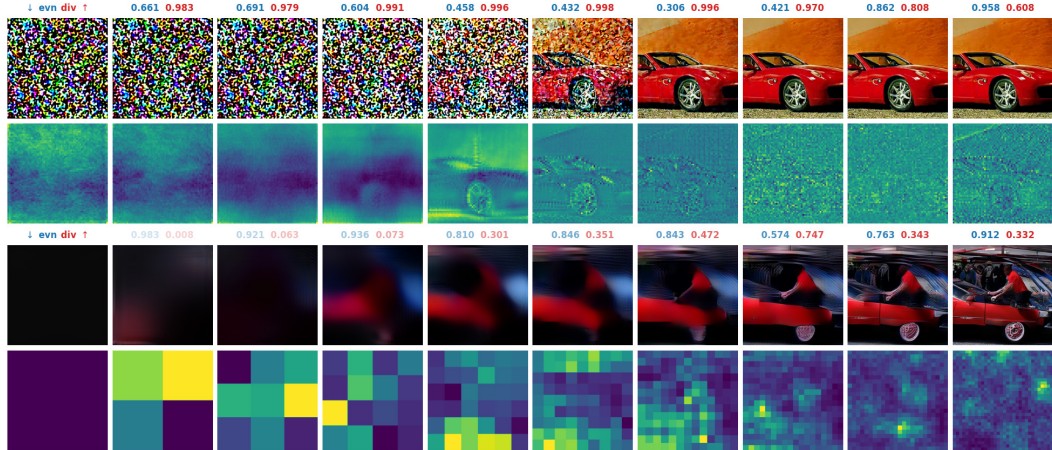

**Figure 2:** Distribution of guidance throughout the sampling process of EDM2 (Karras et al., 2024b) (top) and VAR (Tian et al., 2024) (bottom) on ImageNet $512 \times 512$ class-conditioned generation (class: sports car). For the sake of comparison, the original number of sampling steps of EDM2 (32 steps) has been modified to match VAR. Sampling steps are respectively labelled with their **evenness** and **divergence** scores, with opaqueness corresponding to relative contributions to the weighted mean score. **While EDM2 exhibits guidance signals that are sharp and consistently aligned to foreground objects, VAR exhibits guidance signals that are poorly aligned and becomes progressively fainter.** Additional examples can be found in Appendix A.2, and a visualisation of guidance in EDM2 across the full 32 sampling steps in Appendix A.4.

in diffusion models. Such an effect could be achieved through alternative guidance methods that selectively emphasise certain important regions or foreground features of the image. In light of this, we introduce a new guidance framework based on the key observation from Section 4 that **not all tokens should be equally guided**. As alluded to in the previous paragraph, our method needs to identify semantically important features at inference time and amplify their corresponding guidance signals. In its general form, our framework is defined as

$$\tilde{p}_\theta(s_k|c) = p_\theta(s_k) + \gamma_k \cdot f_k(s_k|c) \cdot \overrightarrow{p_\theta}(s_k|c), \tag{5}$$

We define $f_k : \mathcal{V}^{h_k \times w_k} \to \mathbb{R}^{h_k \times w_k}$ as a function that assigns weights to tokens according to their relative importance. Intuitively, tokens representing the same region tend to carry similar values, and a token surrounded by other salient tokens should itself be considered important. This naturally leads us to the attention mechanism, which is well-suited for capturing such contextual relationships. Prior work has demonstrated that attention is highly effective for selecting informative tokens across diverse vision tasks—including classification Ryoo et al. (2022), captioning You et al. (2016), and in-painting Liu et al. (2019). In particular, Ryoo et al. (2022) showed that pairwise attention can automatically highlight critical visual tokens while maintaining efficiency comparable to state-of-the-art methods. In our framework, however, it is more appropriate to apply attention not to the tokens themselves but to the guidance signals $\overrightarrow{p_\theta}(\cdot|c)$, since these directly steer the sampling process. Accordingly, we realise $f_k$ as a self-attention operation over these guidance signals:

$$f_k(s_k|c) = \text{softmax}\left(\frac{\overrightarrow{p_\theta}(s_k|c)[\overrightarrow{p_\theta}(s_k|c)]^T}{\sqrt{|\mathcal{V}|}}\right). \tag{6}$$

Plugging Equation 6 into Equation 5 yields our proposed guidance scheme, which we term *Information-Grounding Guidance* (IGG).

## 6    EXPERIMENTAL RESULTS

This section presents extensive evaluations of IGG and quantitative analysis on the metrics proposed in Section 4. For class-conditioned image generation, we sampled 50,000 images across 1,000 classes provided by ImageNet (Deng et al., 2009). For text-to-image generation, we conducted experiments on three prompt sets—MJHQ, MS-COCO, and GenEval—sampling 1 image per prompt for the first two set and 4 images per prompt for the last set.

**Table 2:** Comparison of IGG against guidance techniques on ImageNet class-conditioned genera-tion task using representative diffusion (Dhariwal & Nichol, 2021; Rombach et al., 2022; Peebles & Xie, 2023) and SwAR models (Tian et al., 2024). Included guidance methods include *classifier guidance* (CLsG, Dhariwal & Nichol (2021)), *entropy-driven sampling* (EDS, Zheng et al. (2022)), *PixelAsParam* (PxP, Dinh et al. (2023a)), *classifier-free guidance* (CFG, Ho & Salimans (2022)), *progressive guidance* (PROG, Dinh et al. (2023b)), and *representative guidance* (REPG, Dinh et al. (2024)). Wherever possible, we report the guidance schedule scale $w$ (see Section 3). Note that all models but VAR used a fixed guidance schedule. ↑/↓ indicates that higher/lower is better. Best results for each resolution are **bolded**. **Our method** further improves the performance of VAR and achieves new SOTA.

| | Model | Guidance | FID (↓) | IS (↑) | Pre (↑) | Rec (↑) |
|---|---|---|---|---|---|---|
| 256×256 | ADM | CLsG ($w$=1.00) | 4.59 | 186.70 | 0.82 | 0.52 |
| | | EDS | 4.09 | 221.57 | 0.83 | 0.50 |
| | | PxP | 4.00 | 216.11 | 0.81 | 0.53 |
| | | CFG | 3.76 | 191.31 | 0.77 | 0.53 |
| | | PROG | 3.81 | 222.09 | 0.77 | 0.53 |
| | | REPG | 3.34 | 233.26 | 0.85 | 0.52 |
| | LDM | CFG ($w$=1.50) | 3.60 | 246.67 | **0.87** | 0.48 |
| | DiT-XL/2 | CFG ($w$=1.50) | 2.27 | 278.24 | 0.83 | 0.57 |
| | | PROG | 2.25 | 279.36 | 0.82 | 0.58 |
| | | REPG | 2.17 | 268.42 | 0.80 | **0.60** |
| | VAR-$d$30 | CFG ($w$=1.75) | 1.93 | 315.64 | 0.82 | 0.59 |
| | | IGG ($w$=1.85) | **1.92** | **321.28** | 0.82 | 0.59 |
| 512×512 | ADM | CLsG | 7.72 | 172.71 | **0.87** | 0.42 |
| | DiT-XL/2 | CFG | 3.04 | 240.82 | 0.84 | 0.54 |
| | VAR-$d$36 | CFG ($w$=1.50) | 2.61 | 293.7 | 0.82 | 0.56 |
| | | IGG ($w$=2.10) | **2.56** | **314.3** | 0.82 | **0.57** |

## 6.1 CLASS-CONDITIONED IMAGE GENERATION

**Quantitative results.** To assess the scalability of IGG, we apply it to guide the generation of both $256 \times 256$ and $512 \times 512$ images, with VAR Tian et al. (2024) as the backbone model. To further evaluate its off-the-shelf practicality, we directly used pre-trained models provided by the authors on HuggingFace[1]. Our results in Table 2 demonstrate that IGG consistently outperforms CFG on VAR in terms of both FID and IS and achieves a new SOTA amongst guidance methods in both diffusion and SwAR modelling. Furthermore, we observe that the superiority of IGG is more pronounced for $512 \times 512$ image generation. We attribute this phenomenon to the notion that VAR-$d$36 is required to predict larger token maps and, consequently, leaves more room for improvement regarding the task of concentrating on semantically important tokens. Indeed, this notion is supported by the fact that the evenness and divergence scores for VAR-$d$36 are worse than those of VAR-$d$30.

**Qualitative results.** We perform side-by-side visual comparisons between CFG and our method similar to our analysis in Section 4. Figure 1 depicts a representative comparison. In numerous cases, IGG gathers guidance signals towards semantically important tokens and forms contours clearly resembling the corresponding foreground objects, mimicking the patterns of CFG in dif-fusion modelling (Figure 2). This finding, along with the improvement of our method over CFG, further reinforces the notion that the superiority of diffusion models can be explained by the insights presented in Section 4.

## 6.2 TEXT-TO-IMAGE GENERATION

**Quantitative results.** To test the effectiveness of IGG on SwAR models for text-to-image tasks, we compare IGG against CFG on the MJHQ (Li et al., 2024a), MS-COCO (Lin et al., 2014), and GenEval (Ghosh et al., 2023) benchmarks. Two recent SwAR backbones, VAR-CLIP (Zhang et al., 2024) and Switti (Voronov et al., 2025), were considered for IGG, while we also include baselines for other popular backbones (SDXL (Podell et al., 2023), LlamaGen (Sun et al., 2024), and HART

---

[1]https://huggingface.co/FoundationVision/var.

**Table 3:** Comparison of IGG against classifier-free guidance (CFG) on various text-to-image benchmarks using VAR-CLIP and Switti. Popular baselines using CFG are also included.↑/↓ indicates that higher/lower is better. Best results for each resolution are **bolded**. Our method demonstrates **strong overall performance across all benchmarks.**

| Resolution | Model | Guidance | GenEval Overall (↑) | MJHQ-30K FID (↓) | CLIP (↑) | PickScore (↑) | IR (↑) | COCO-30K FID (↓) | CLIP (↑) | PickScore (↑) | IR (↑) |
|---|---|---|---|---|---|---|---|---|---|---|---|
| $256^2$ | VAR-CLIP | CFG | 0.22 | 32.95 | 0.184 | 0.177 | -1.78 | 10.95 | **0.264** | **0.198** | **-0.87** |
| | | IGG | **0.23** | **32.27** | **0.221** | **0.180** | **-1.58** | **10.93** | **0.264** | **0.198** | -0.89 |
| $512^2$ | SDXL | CFG | 0.55 | 7.6 | 0.384 | 0.217 | 0.78 | 14.4 | 0.360 | 0.226 | 0.77 |
| | LlamaGen | CFG | 0.32 | 26.9 | 0.288 | 0.194 | -0.45 | 44.8 | 0.274 | 0.208 | -0.25 |
| | HART | CFG | 0.55 | 5.8 | 0.366 | 0.216 | 0.84 | 20.9 | 0.341 | 0.223 | 0.75 |
| | Switti | CFG | 0.62 | 9.5 | 0.388 | **0.221** | **1.15** | 17.6 | 0.355 | **0.228** | 0.93 |
| | | IGG | **0.64** | **7.09** | **0.389** | 0.220 | 1.14 | **16.9** | **0.357** | **0.228** | **0.97** |

(Tang et al., 2024)) for ease of comparison. The metrics to compare were GenEval, FID, CLIP, PickScore (Kirstain et al., 2023), and ImageReward (IR) (Xu et al., 2023). As shown in Table 3, IGG helps Switti to achieve significant improvement on different metrics, especially Geneval and FID. These improvements translate to stronger complex prompt modelling capacity and arguably enhance image quality. Reasoning for these improvements similar to class-conditioned generation, since the misalignment property is relaxed, images generated under IGG received more guidance on semantically important regions without introducing more text-conditioned artefacts in the less important regions.

**Qualitative results** Sample generations from Switti were shown in Figure 3 for comparison between the three guidance schemes: No guidance, CFG, and IGG. Observing the samples, we see that IGG achieve the best generation overall. With the no guidance scheme, due to receiving less text conditioning, the generation became too simple, with incomplete or prompt-disobedient objects. For CFG, while the objects were more complex, each patch receiving equal text-conditioning introduces artefacts to the generations. IGG, on the other hand, alternating guidance in regions of the image, resulted in less artefact-prone, more prompt-following and complex generations. This behaviour follows what we observed in the quantitative results, affirming the efficacy of our method.

## 6.3 METRIC ANALYSIS

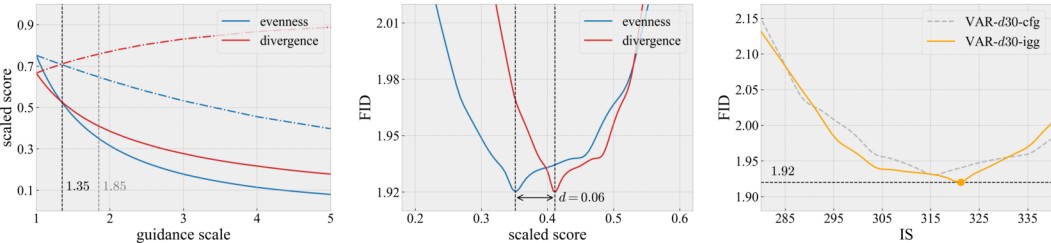

**Figure 4:** Analysing the relationships between various metric scores attained by VAR-$d$30-IGG. From left to right: effect of changing guidance scales on reported evenness and divergence scores, where dashed and solid lines depict raw and scaled scores respectively; correspondence between FID and scaled evenness and divergence scores; and FID-IS trade-off curve.

**Guidance scale vs. evenness and divergence.** To better understand the dynamics between guidance weight $w$ and the evenness and divergence metrics detailed in Section 4, we computed these metrics over an extensive range of guidance weights on VAR-$d$30-IGG. However, we note that there is an inherent correlation between each metric with $w$ since the nudges they evaluate include the guidance scales $\gamma_k$ (see Equation 2), whose values depend on $w$ by design. Thus, to disassociate these metrics, we scaled them by a factor corresponding to the (reciprocal of the) respective guidance scales applied. Both the original scores (dashed lines) and scaled scores (solid lines) were plotted in Figure 4 (left). Interestingly, evenness and divergence exhibit a similar trend: they improve with increasing $w$, at a decaying rate proportional to $w$. Notably, these two scores meet at $w = 1.35$ (black line). Evaluating VAR-$d$30-IGG at this guidance weight yielded an FID of $\approx 1.98$, which is not too far off the optimal FID obtained at $w = 1.85$ (grey line). This result demonstrates the potential

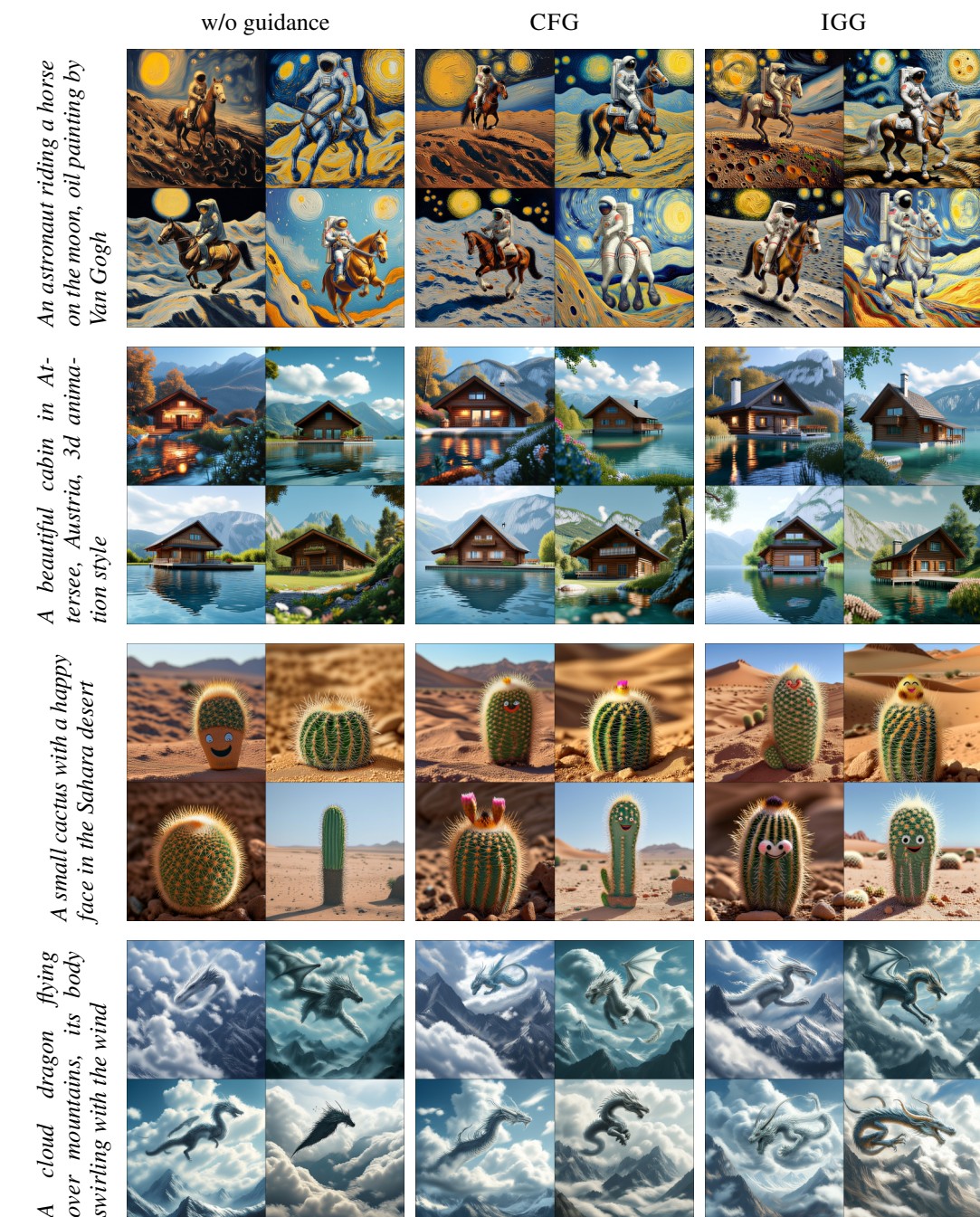

| | w/o guidance | CFG | IGG |

**Figure 3:** Example $1024 \times 1024$ generations of Switti under three guidance schemes: no guidance, CFG, and IGG (ours). Without IGG, CFG or vanilla sampling of Switti has higher chance of generating failure features, around one in four samples.

of evenness and divergence to approximate the optimal guidance weight to apply to a model, which could significantly alleviate the labour of hyperparameter tuning for large models.

**Evenness and divergence vs. FID.** We also investigated the relationship between the proposed metrics and FID, as depicted in Figure 4 (centre). The similarity between the (scaled) scores is once again reflected. In particular, they both exhibit a sharp FID curve. Notably, the optimal FID is achieved at almost-equal evenness and divergence (within 0.06 away from each other). This indicates a close connection between evenness-divergence equilibrium and FID optimality, providing further support for the notion raised in the above analysis.

**FID vs. IS.** Figure 4 (right) depicts the trade-off between sample diversity (IS) and fidelity (FID) of our method compared to CFG. As IS increases, both methods initially achieve lower FID, but beyond IS of $\approx 320$, further increases in diversity come at the cost of worsening fidelity, suggesting a natural limit to the achievable balance. Notably, within the proximity of the optimal trade-off, our method consistently outperforms CFG. Overall, the curve underscores that our method provides a slightly more favourable balance between diversity and realism. As suggested in Table 2, we expect this advantage to increase with model scale.

## 6.4 ABLATION STUDY

**Table 4:** Ablation study on VAR-$d$30-IGG. Each row (except the first row) presents the results of a single modification to the original implementation (first row). $\uparrow/\downarrow$ indicates that higher/lower is better. Best results are **bolded** and second-best results are underlined.

| Description | FID ($\downarrow$) | IS ($\uparrow$) | Pre ($\uparrow$) | Rec ($\uparrow$) | Evn ($\downarrow$) | Div ($\uparrow$) |
|---|---|---|---|---|---|---|
| Vanilla      (*no changes*) | **1.92** | 321.2 | 0.82 | 0.59 | 0.646 | 0.757 |
| Mixed scheme  ($w=w'$=0.75) | 5.48 | **415.6** | **0.88** | 0.48 | **0.492** | **0.827** |
| Fixed schedule    ($\gamma_k$=1.85) | 4.29 | 359.5 | 0.87 | 0.50 | 0.727 | 0.684 |
| Sliding window ($s_k=\sqrt{h_k w_k}$) | **1.92** | 321.5 | 0.82 | **0.60** | 0.608 | 0.759 |

**Mixed guidance schemes.** Following the success of mixing CFG and IGG in text-to-image generation, we wanted to investigate its effect on other evaluation metrics. In our experiment, we used an equal guidance weight of $0.75$ for both the CFG and IGG components (Equation 7). Although the resulting model was able to improve upon evenness and divergence, it fell short in terms of FID. This discrepancy emphasises the importance of achieving an equilibrium between evenness and divergence as opposed to arbitrarily improving them independently, as revealed in Section 6.3. It also, once again, calls for further investigation on the reason why mixing guidance schemes works so well in diffusion modelling.

**Fixed guidance schedule.** To assess the importance of choosing the right guidance schedule, we replaced the default schedule in Tian et al. (2024) with a fixed schedule. Specifically, we set the guidance scale $\gamma_k$ at every scale level to $1.85$. This change turns out to be detrimental to the model's performance, with a worse FID, evenness, and divergence compared to the vanilla implementation. This result is a testament to the major role that the ratio-based guidance schedule plays in the efficacy of IGG and, quite likely, also CFG.

**Sliding-window guidance.** The original implementation of IGG performs a global attention computation at each scale level (see Equation 6). Inspired by the successes of localised attention mechanism in NLP (Beltagy et al., 2020), we implemented a variant of IGG which utilises a *scale-wise 2-D sliding window* for attention computation, where the size $s_k$ of the sliding window is scaled accordingly at step $k$. In our experiment, we set $s_k := \sqrt{h_k w_k}$. The result of the experiment does not indicate any non-negligible improvement from the original implementation. Despite this, we expect that the computational cost saved through using localised attention will become greatly beneficial for sampling high-resolution images (e.g., $1024 \times 1024$), since the model will then likely need to predict more token maps at greater sizes.

## 7 CONCLUSION

In this work, we investigated the dynamics of guidance in scale-wise autoregressive (SwAR) models and revealed a key limitation: unlike diffusion models, guidance in SwAR is often dispersed and misaligned, weakening semantic consistency. Building on this insight, we introduced a novel guidance framework that aims to anchor guidance signals to semantically important tokens via contextual attention, and propose a realisation in Information-Grounding Guidance (IGG). Our experiments across both class-conditioned and text-to-image generation tasks demonstrated that IGG consistently improves fidelity, coherence, and alignment over classifier-free guidance, while also providing interpretable metrics that correlate with sampling quality. Beyond immediate performance gains, our analysis highlights the broader principle that not all tokens should be guided equally, opening avenues for future research on new guidance strategies in autoregressive generative modelling by utilising our proposed framework.

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

# A    APPENDIX

## A.1    DIVERGENCE EVALUATION PROCEDURE

Algorithm 1 details the procedure for systematically computing the divergence score as defined in Section 4.

---

**Algorithm 1** Procedure for computing divergence score.

---

**Require:** Sampled image $x$ and associated token maps $s := (s_1, \cdots, s_K)$ of sizes $\{h_k, w_k\}_{k=1}^K$.
**Ensure:** Divergence score for $x$.

  1: $M := \text{segment}(x)$                                         ▷ Obtain binary segmentation mask
  2: $J := 0$
  3: **for** $k \in 2..K$ **do**                                       ▷ Skip $s_1$ (see Section 4)
  4:      $M_k := \text{interpolate}(M, h_k, w_k)$            ▷ Down-sample mask to $h_k \times w_k$
  5:      $\vec{p_k} := \vec{p_\theta}(s_k|c)$                      ▷ Obtain guided nudge distribution
  6:      $s_k^* := \text{sample}(s_k[\neg M_k], h_k, w_k)$     ▷ Sample with replacement $h_k w_k$ unguided tokens
  7:      $\vec{q_k} := \vec{p_\theta}(s_k^*|c)$                     ▷ Obtain unguided nudge distribution
  8:      $J := J + h_k w_k / \sum_{i=2}^K h_i w_i \cdot \text{divergence}(\vec{p_k}, \vec{q_k})$   ▷ Weighted mean analogous to PEI
  9: **end for**
10: **return** $J$

---

## A.2    COMPARING GUIDANCE IN DIFFUSION AND SWAR MODELS

Figure 5 presents additional side-by-side comparisons of the guidance signals observed during the sampling processes of EDM2 (Karras et al., 2024b) and VAR (Tian et al., 2024). These visualisations make it easier to contrast how guidance is distributed in diffusion models compared to SwAR models. In particular, EDM2 exhibits relatively stable and balanced guidance, while VAR tends to experience progressively weaker signals across sampling steps. This distinction provides further evidence for our hypothesis that the uneven distribution of guidance contributes to the sampling quality gap between the two model families.

## A.3    COMPARING CFG AND IGG

In Figure 6, we present additional results comparing samples generated with CFG against those obtained using IGG. These examples highlight the qualitative differences between the two approaches, particularly in terms of semantic coherence and visual fidelity. While CFG often struggles to maintain consistency across fine-grained details, IGG demonstrates a stronger ability to emphasise and preserve semantically important features. These examples further illustrate how our method balances guidance strength without sacrificing diversity in the generated outputs.

## A.4    GUIDANCE IN DIFFUSION MODELS ACROSS ALL TIMESTEPS

Figure 7 visualises the full sampling process of EDM2 (Karras et al., 2024b). Interestingly, guidance signals seem to be the most pronounced in the middle of the sampling process, suggesting that the role played by guidance is the most influential around this period. This is consistent with the finding of Kynkäänniemi et al. (2024), providing further support for our proposed interpretation of CFG.

## A.5    APPLYING IGG TO DIFFUSION MODELS

Although IGG is tailored for SWaR modelling, we also evaluated it in diffusion modelling to assess its generalisability and potential to be applied in this setting. To this end, we introduce a guidance scheme that is a mixture of CFG and IGG,

$$\tilde{p}_\theta^{\text{MIX}}(s_k|c) = p_\theta(s_k) + \gamma_k\, \tilde{p}_\theta^{\text{CFG}}(s_k|c) + \gamma_k'\, \tilde{p}_\theta^{\text{IGG}}(s_k|c), \tag{7}$$

where $p_\theta^{\text{CFG}}(\cdot|c)$ is the guidance scheme defined in Equation 2. Consequently, this mixed guidance scheme requires a pair guidance weights $(w, w')$ to define the guidance schedules $\{\gamma_k, \gamma_k'\}$. In this experiment, we used EDM2 (Karras et al., 2024b), also pre-trained, as our backbone model. We opted for a *fixed* guidance schedule (that is, $\gamma_k = w, \gamma_k' = w'$) to enable direct comparison with

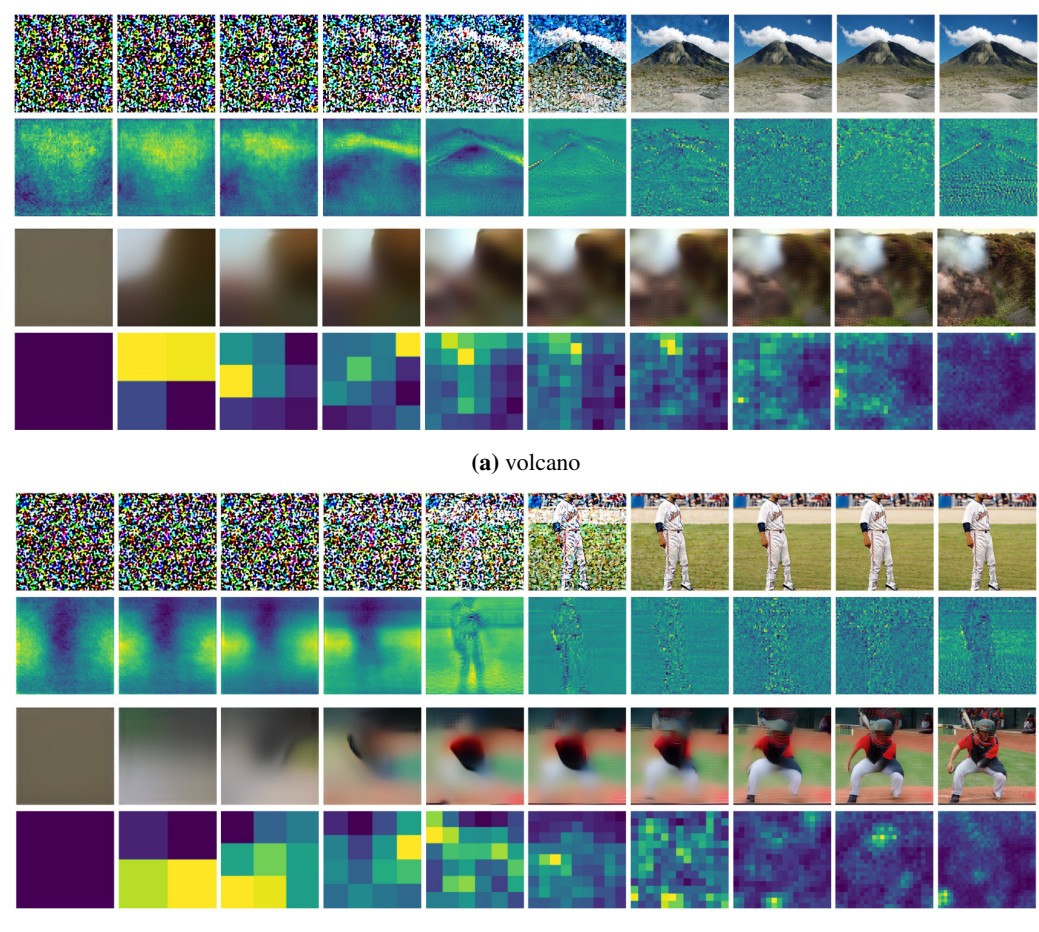

**(a)** volcano

**(b)** baseball player

**Figure 5:** Additional comparisons of guidance signals in EDM2 (Karras et al., 2024b) (top) and VAR (Tian et al., 2024) (bottom).

**Table 5:** Comparison of IGG against classifier-free guidance (CFG) on class-conditioned generation task using pre-trained EDM2 (Karras et al., 2024b). $w$ is the guidance weight used to define the guidance schedule (see Section 3). ↑/↓ indicates that higher/lower is better. Best results are **bolded**.

| Model | Guidance | | FID ($\downarrow$) |
|---|---|---|---|
| EDM2-S | CFG | ($w$=1.40) | 2.29 |
| (512×512) | CFG-IGG ($w,w'$=1.40,−0.40) | | **2.20** |
| EDM2-XXL | CFG | ($w$=1.20) | 1.84 |
| (512×512) | CFG-IGG ($w,w'$=1.20,−0.20) | | **1.82** |

the original CFG implementation in EDM2, which also used a fixed schedule. We found that by keeping $w$ at the optimal value for vanilla CFG (as reported in Kynkäänniemi et al. (2024)) and setting $w'$ to a *negative* value improves the sampling quality of EDM2 (see Table 5), while setting it to a positive value does not. We posit that this observation stems from the "denoising" nature of diffusion modelling, namely, DDPMs (Ho et al., 2020; Song et al., 2020). In this context, it is possible that removing some noise from semantically important regions leads to an acceleration in the denoising process for those regions and, as a result, allows time for further refinements on areas with high levels of detail in the fixed sampling time-frame. Nevertheless, we emphasise that this hypothesis is based purely on empirical observations and that its validity would greatly benefit from further research.

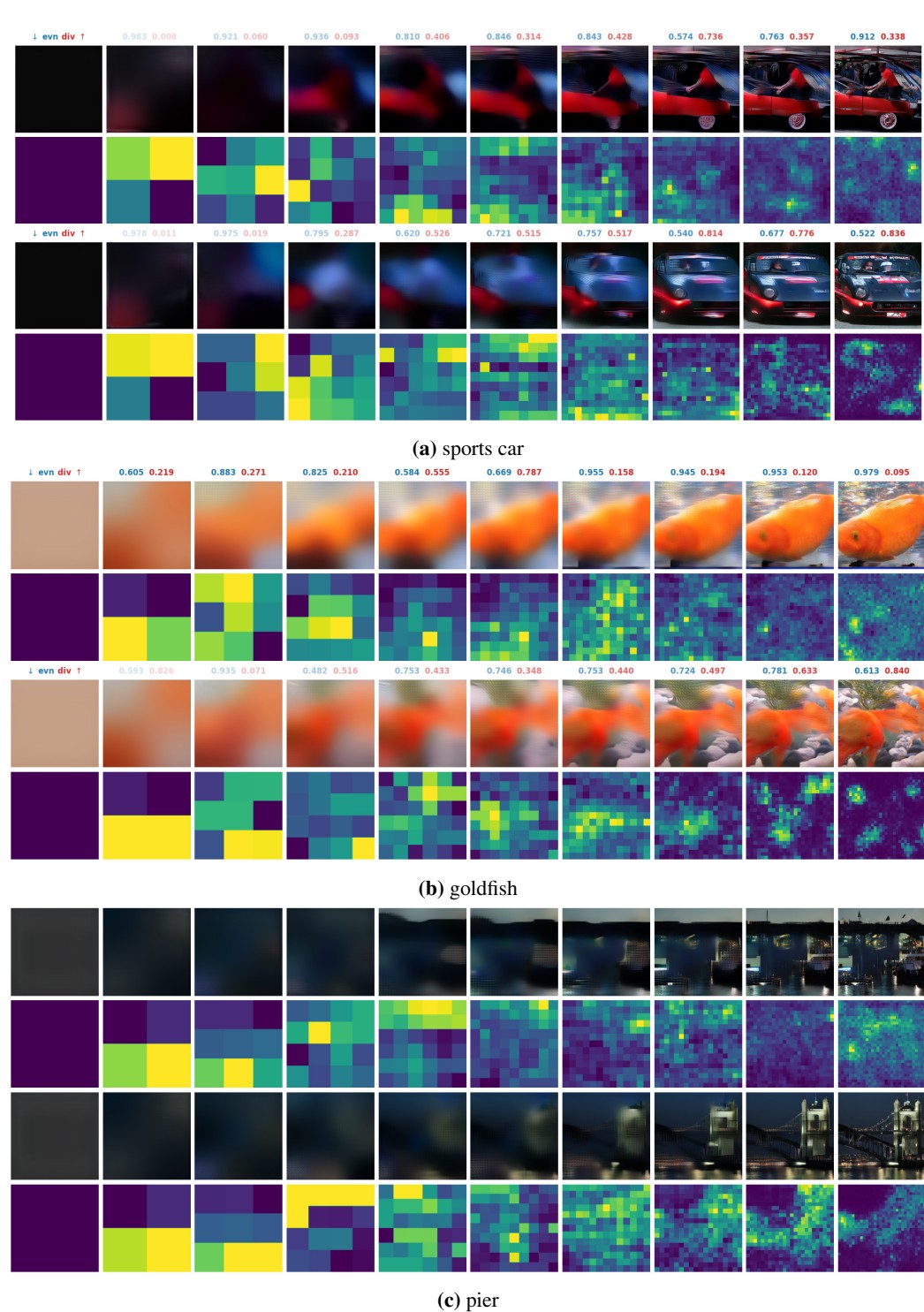

**Figure 6:** Additional side-by-side comparisons of sampling in VAR-*d*36 (Tian et al., 2024) using CFG and IGG.

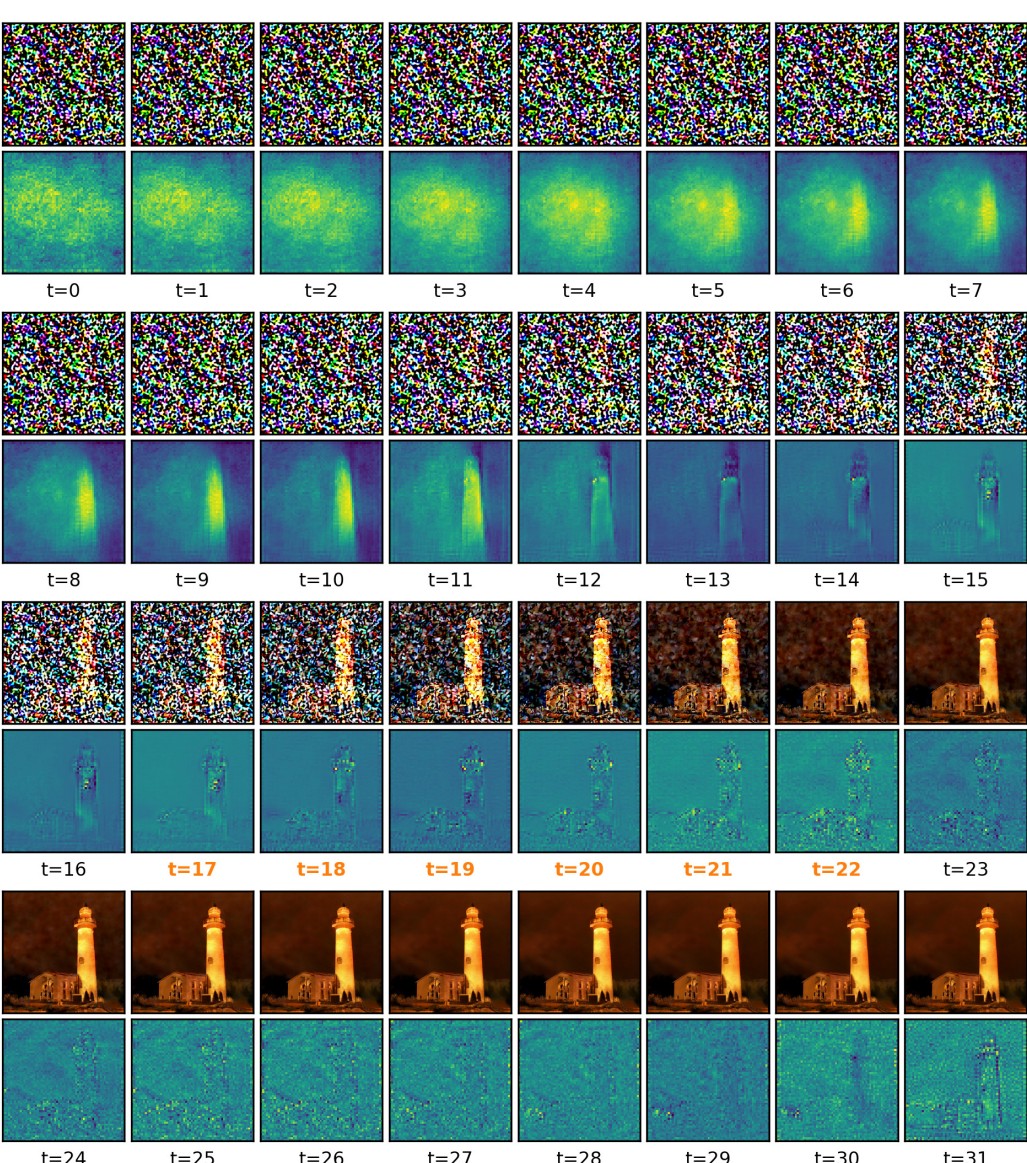

**Figure 7:** Visualisation of guidance signals for every timestep in the original sampling process of EDM2 (Karras et al., 2024b). Timesteps falling within the guidance interval reported in Kynkäänniemi et al. (2024) are highlighted in orange. This interval coincides with the transition from noise to image, which is where guidance plays the most influential role.

