# OpenReview forum: "Not All Tokens are Guided Equal: Elucidating Guidance in Autoregressive Modelling"
_ICLR.cc/2026/Conference — Submitted to ICLR 2026_

### Official Review · Reviewer_7N6s · 2025-10-26

**Soundness:** 3
**Presentation:** 2
**Contribution:** 3
**Rating:** 6
**Confidence:** 4

**Summary:**

Classifier-free guidance (CFG) enhances conditional diffusion models by interpolating between unconditioned and conditioned predictions. However, the authors argue that a naive application of CFG fails to yield satisfactory results in scale-wise autoregressive (SwAR) models, where prediction occurs across scales. They argue that unlike diffusion models, guidance in SwAR does not account for the semantic relationships between tokens, a limitation the authors quantify using two newly proposed metrics. To address this, they introduce Information-Grounding Guidance (IGG), a method that anchors the guidance signal to semantically important regions via attention. Their experiments demonstrate the effectiveness of IGG across both class-conditioning and text-conditioning benchmarks.

**Strengths:**

1 - I think the authors raise an important question: why should we assume that classifier-free guidance (CFG), originally developed for diffusion models, will work out of the box for autoregressive models, specifically, scale-wise autoregressive ones?

2 - I appreciate that the authors pinpoint the issue through analyses of guidance token evenness and divergence, and further reinforce their intuition with extensive visualizations.

3 - The method is remarkably simple and incurs almost no additional cost compared to CFG, essentially just an extra dot product between tokens.

**Weaknesses:**

My comments are intended more as questions than as criticisms, so I will include them in the “Questions” section.

**Questions:**

Conceptually, the authors’ arguments are sound, and the visual results clearly illustrate the effect of their intervention. What concerns me, however, is its impact on final performance.

1 - While the authors claim that the performance gap becomes more pronounced at higher resolutions—as shown in Table 2 (ImageNet 512×512)—this trend does not appear in the text-to-image benchmark (Table 3). In particular, the gap for Switti is actually smaller than for VAR-CLIP, which seems inconsistent with their stated hypothesis.

2 - In Section 6.3, the authors emphasize the importance of maintaining a balance between evenness and divergence, rather than optimizing each independently, to achieve the best outcomes. If this holds true, I’m curious how they explain the pronounced imbalance between these two factors in diffusion models compared to SwAR, as shown in Table 1.

---

> ### Author Response · Authors · 2025-11-26
> **Response to Reviewer 7N6s**
>
> We are grateful to Reviewer 7N6s for their appreciation towards our work. Regarding their questions:
>
> ### **Q1. Is IGG more potent at higher resolutions?**
>
> We assert that the validity of the claim regarding IGG's efficacy at high resolution is not affected by the results in Table 3. Indeed, in Table 2 (ImageNet512), the models being compared are simply two configurations of the same architecture (VAR), which only differ in the number of transformer blocks. On the other hand, **there are fundamental differences between Switti and VAR-CLIP**, including but not limited to:
>
> - Usage of text encoder (VAR-CLIP uses a single encoder while Switti concatenates multiple encoders);
> - Backbone architecture (VAR-CLIP uses a causal transformer while Switti uses a non-causal transformer); and
> - Training dataset (VAR-CLIP was trained on ~1M examples while Switti was trained on ~100M examples).
>
> Due to these confounding differences, naively comparing the performance gaps induced by IGG between VAR-CLIP and Switti is therefore misleading.
>
> ### **Q2. How important is maintaining evenness-divergence balance?**
>
> Balance between evenness and divergence is indeed a factor to be considered when optimising the performance of guidance methods, but **it is certainly neither a sufficient nor the most important factor**. In particular, before the reduction of the evenness-divergence gap can take place, one needs to independently optimise these metrics to be within a desirable threshold.
>
> Once these thresholds have been reached (in the case of diffusion models), the effect of balancing evenness and divergence becomes apparent when we compare the results of EDM2-S and EDM2-XXL. As reported in Table R6 (which was extracted from Table 1), the evenness-divergence gap is narrower in the latter, and this, indeed, translated to better performance in practice.
>
> | Model    | Evenness | Divergence | Gap          |
> | -------- | -------- | ---------- | ------------ |
> | EDM2-S   | 0.486    | 0.983      | 0.497        |
> | EDM2-XXL | 0.560    | 0.964      | 0.404 (-0.093) |
>
> *Table R6. Evenness-divergence gap in diffusion models. Moving from EDM2-S to EDM2-XXL results in a 19% decrease in gap width.*
>
> We hope these answers have sufficiently addressed the reviewer's questions. Please do not hesitate to follow up for any further enquiries.

---

### Official Review · Reviewer_sSUr · 2025-10-30

**Soundness:** 2
**Presentation:** 3
**Contribution:** 3
**Rating:** 6
**Confidence:** 4

**Summary:**

The paper studies why classifier-free guidance (CFG) behaves poorly in scale-wise autoregressive (SwAR) image generators and introduces Information-Grounding Guidance (IGG), which reweights guidance across tokens using self-attention over the guidance “nudges.” IGG is intended to concentrate guidance on semantically salient regions, aligning content and conditioning more tightly, and is shown to improve FID/IS on ImageNet class-conditional generation and several text-to-image benchmarks.

**Strengths:**

- Clear diagnosis: the paper quantifies that SwAR guidance is uneven and often misaligned compared to diffusion, using evenness (PEI) and divergence (JSD) metrics and qualitative visualizations.
- Method is simple and principled: IGG modifies CFG by multiplying the guidance signal with an attention-derived token-importance map, with a precise formulation (Eq. 5–6).
- Useful analysis/ablations: the paper connects guidance scale schedules to evenness/divergence and shows mixed CFG+IGG schemes can improve metrics but harm FID, arguing for balance rather than one-sided optimization.

**Weaknesses:**

- Magnitude of gains on class-conditional ImageNet is modest in places (e.g., VAR-d30 FID 1.93→1.92, Pre/Rec almost no gain), raising questions about practical significance at lower scales or resolutions.
- Compute concerns: IGG applies global self-attention over guidance at each scale; authors suggest a sliding-window variant mainly for future efficiency at high resolutions, implying the baseline IGG may be costlier as resolution grows. Evidence of actual runtime/memory overhead is not reported.
- Sensitivity and tuning: while the paper discusses schedules and scale, more systematic reporting on sensitivity to the guidance weight \omega and the stability of results across seeds/backbones would strengthen claims.

**Questions:**

- It seems that when comparing CFG and IGG, the parameter \omega is set differently. Please explain the reason.
- The authors use the notation (p) to denote both the model and its parameters, which may hinder readability; this is not recommended.
- Have you tested IGG with non-VAR SwAR/tokenizers (e.g., different codebooks or continuous tokens) to assess architectural dependence?
- For text-to-image, can you report human-eval or instruction-following error rates alongside CLIP/PickScore/IR to validate that IGG’s regional emphasis improves semantic faithfulness, not just aesthetics?

---

> ### Author Response · Authors · 2025-11-26
> **Response to Reviewer sSUr [1/2]**
>
> We sincerely thank Reviewer sSur for taking their time to review our manuscript. Regarding their concerns:
>
> ### **1. IGG is performant even at lower scales**
>
> While FID gains at lower resolutions are less significant, we emphasise that the **practical advantage of IGG lies in its scalability and superior diversity-fidelity trade-off**. Indeed, IGG yielded a marked increase in IS, namely from 315.64 to 321.28, for VAR-d30 (without sacrificing FID), thereby achieving a more optimal trade-off than CFG.
>
> We further emphasise that our method was designed to target the misalignment of guidance across scales, a problem inherently less severe at lower resolutions. Thus, modest gains at 256x256 resolution are expected. Meanwhile, the substantial performance leaps at 512x512 resolution indicate that **IGG effectively alleviates the misalignment problem at higher resolutions, where CFG typically fails**.
>
> ### **2. IGG is efficient even at high resolutions**
>
> Regarding the suggestion about adopting a sliding-window attention mechanism for IGG, we clarify that this is merely a remark on the flexibility of our method. Notwithstanding this remark, throughout our experiments at both 256x256 and 512x512 resolutions, we found that **IGG incurs negligible overhead compared to CFG**. To consolidate this observation, we report the inference time and GFLOP counts of CFG vs. IGG in Table R1.
>
> | Model         | Inference time (per image)  | GFLOPs (per image) |
> | ------------- | --------------------------- | ------------------ |
> | VAR-d30-*CFG* | 0.1391                      | 259.93             |
> | VAR-d30-*IGG* | 0.1408 (+0.0017)            | 259.93 (+0.00)     |
> | VAR-d36-*CFG* | 0.7009                      | 311.12             |
> | VAR-d36-*IGG* | 0.7103 (+0.0094)            | 311.12 (+0.00)     |
>
> *Table R1. Inference time and GFLOP counts. Compared to CFG, IGG incurs a mere 1% increase in inference time and no additional FLOPs.*
>
> ### **3. IGG exhibits similar sensitivity and stability to CFG**
>
> ### 3.1. Sensitivity to guidance weight
>
> Regarding the concern about the sensitivity of IGG to the guidance weight, we refer the reviewer to Figure 4 (right) in the manuscript. This figure depicts the results in FID (y-axis) and IS (x-axis) yielded by CFG (grey) vs. IGG (yellow) across the same range of guidance weights, namely from $w$=1.0 to $w$=2.5. Apparently, our method exhibits a similar trend compared to CFG, indicating comparable sensitivity to $w$.
>
> ### 3.2. Stability of results
>
> To address the concern about the stability of results, we ran the class-conditioned generation experiment (Table 2) of VAR-d30 across 10 different seeds. Table R3 reports the standard errors for each metric.
>
> | Method | FID   | IS    | Prec  | Rec   |
> | ------ | ----- | ----- | ----- | ----- |
> | CFG    | 0.005 | 0.538 | 0.000 | 0.002 |
> | IGG    | 0.009 | 0.627 | 0.000 | 0.002 |
>
> *Table R3. Standard errors of CFG and IGG in class-conditioned generation using VAR-d30.*
>
> These results indicate that **CFG and IGG exhibit similar variability across all metrics**. To verify this hypothesis, an additional F-test was conducted. The results are reported in Table R4. Since all p-values fall above 0.05, we concluded that the differences in variance between the two methods are not statistically significant.
>
> | Metric | F-statistic | p-value | p<0.05 |
> | ------ | ----------- | ------- | ------ |
> | FID    | 3.190       | 0.099   | False  |
> | IS     | 1.358       | 0.656   | False  |
> | Prec   | n/a         | 1.000   | False  |
> | Rec    | 2.095       | 0.286   | False  |
>
> *Table R4. Comparison of variances of CFG and IGG via an F-test. p<0.05 indicates statistical significance.*

---

> ### Author Response · Authors · 2025-11-26
> **Response to Reviewer sSUr [2/2]**
>
> Regarding the reviewer's questions:
>
> ### **Q1. Why is the guidance weight set differently for CFG and IGG?**
>
> The difference in the guidance weight $w$ set for CFG and IGG stems from the fact that IGG already applies an adaptive nudge on each token (from its self-attention value). Thus, the role of $w$ in our method is simply to tune the magnitudes of these nudges. CFG, on the other hand, applies a constant nudge for all tokens, and the magnitude of this nudge is completely determined by $w$. In this light, it is natural that the optimal $w$ for each method is different.
>
> ### **Q2. Overloading notation $p$ hinders readability?**
>
> We thank the reviewer for pointing this out. In hindsight, we agree that overloading the notation $p$ may lead to confusion and will take this into consideration when revising our manuscript.
>
> ### **Q3. Can IGG be applied to non-SwAR architectures?**
>
> We note that IGG is originally designed to tackle the misalignment of guidance that gradually arises as the model progresses from one scale to the next, a problem most relevant for SwAR models.
>
> Nevertheless, **IGG can indeed be extended to general AR models**. To verify this claim, we implemented our method on LlamaGen, a traditional token-wise model. To adapt to its next-token prediction approach, we modified the original formulation (see Section 5 in the manuscript) to causal attention, that is, every token attends only to the past tokens up to a set window size $s$. In our experiment, we set $s=1$ for simplicity.
>
> As reported in Table R5, compared to CFG, **IGG was able to attain a 2% improvement in FID**. Notably, it did so without worsening IS (even slightly improving it), indicating a more optimal quality-diversity trade-off.
>
> | Model                     | FID      | IS         | Prec     | Rec      |
> | ------------------------- | -------- | ---------- | -------- | -------- |
> | LlamaGen-L-*CFG* (w=2.00) | 3.07     | 256.06     | **0.83** | 0.52     |
> | LlamaGen-L-*IGG* (w=2.00) | **3.01** | **256.42** | **0.83** | **0.53** |
>
> *Table R5. Comparison of IGG against CFG on class-conditioned generation for ImageNet256 with LlamaGen-L as the backbone.*
>
> ### **Q4. Does IGG improve semantic faithfulness?**
>
> While we are unable to report human-eval metrics due to budget constraints, we can report the full results of GenEval; see table below. For reference, GenEval (https://arxiv.org/pdf/2310.11513) is an object-focused framework which enables the automated evaluation of text-to-image models with strong human agreement.
>
> | Model          | Overall  | Single-object | Two-object | Counting  | Colors | Position  | Color attribution |
> | -------------- | -------- | ------------- | ---------- | --------- | ------ | --------- | ----------------- |
> | VAR-CLIP-*CFG* | 0.22     | 0.694         | 0.068      | **0.113** | 0.431    | 0.003     | 0.003             |
> | VAR-CLIP-*IGG* | **0.23** | **0.716**     | **0.078**  | 0.094     | **0.452**  | **0.008** | **0.005**         |
> | Switti-*CFG*   | 0.62     | 0.997         | 0.753      | 0.556     | 0.883    | 0.113     | 0.398             |
> | Switti-*IGG*   | **0.64** | **1.000**      | **0.808**  | **0.609** | **0.888**  | **0.123** | **0.430**         |
>
> *Table R2. Full evaluation of IGG on GenEval. VAR-CLIP generates 256x256 images while Switti generates 512x512 images.*
>
> Notably, **IGG consistently outperforms CFG in almost every metric**, with the only exception of the Counting score for VAR-CLIP. In particular, the biggest improvement was found in the Two-object score, where a 15% and 7% increase was attained for VAR-CLIP and Switti, respectively. This result **highlights IGG's ability to ground important semantic information even in complex settings** where there are multiple foreground objects.
>
> We hope these answers have sufficiently addressed the reviewer's concerns and questions. Please do not hesitate to follow up for any further enquiries.

---

### Official Review · Reviewer_ksrD · 2025-10-31

**Soundness:** 3
**Presentation:** 2
**Contribution:** 2
**Rating:** 2
**Confidence:** 4

**Summary:**

This paper proposes  Information-Grounding Guidance (IGG), a novel guidance framework for AR image generation models. Unlike classifier-free guidance (CFG), which uniformly applies guidance across all tokens, IGG introduces a token-specific guidance mechanism that dynamically adjusts the strength based on the semantic importance of each token. IGG is evaluated on class-conditional and text-to-image tasks, showing improvements in fidelity, coherence, and alignment over CFG.

**Strengths:**

1. Conceptual Innovation: The core insight—that not all tokens should be guided equally—is both intuitive and impactful. The proposal of token-level, interpretable guidance addresses a fundamental limitation of uniform CFG in AR models.
2. Rich experiments: Experiments have been conducted on multiple datasets for verification, and the results are convincing.

**Weaknesses:**

1. The improvement in quantitative indicators brought by IGG is insufficient, and it seems that in the visualization results provided in the paper, it is also difficult to see its obvious advantages over CFG
2. Computational Overhead and Efficiency Concerns: IGG introduces an auxiliary network and additional inference steps to compute token weights. The paper does not report latency or FLOPs comparison with CFG. A small overhead per token could significantly impact practical usability. This is a critical omission for a method claiming to improve sampling.
3. As a training-free guidance optimization method, merely verifying its effectiveness on the scale-wise autoregressive method cannot prove its universality. The authors should conduct experimental verification on more types of AR models.
4. I think the explanation of the core method IGG in Section 5 is a bit crude and not conducive to understanding its working principle.

**Questions:**

1. What is the computational cost of IGG per sampling step?
2. Can IGG be applied to naive autoregressive models, such as LlamaGen?

---

> ### Author Response · Authors · 2025-11-26
> **Response to Reviewer ksrD [1/2]**
>
> We sincerely thank Reviewer ksrD for their insightful feedback. Regarding their concerns:
>
> ### **1. IGG considerably outperforms CFG**
>
> We claim that the improvements attained by IGG are sufficient to justify its practicality. In fact, **we were able to outperform CFG in almost all experiments**, with the only exception being class-conditioned generation on VAR-d30 (where we, nevertheless, matched the baseline). Notably:
>
> - In class-conditioned generation, **IGG yielded a 2% and 7% improvement** in FID and IS respectively on VAR-d36 for ImageNet512, see Table 2; and
> - In text-to-image generation, **IGG yielded a 3% and 25% improvement** in GenEval (Overall) and FID (for MJHQ-30K), respectively, on Switti, see Table 3.
>
> Given the already impressive results attained by these models prior to our work and the simplicity of the proposed methodology, there are concrete grounds to argue for IGG's practicality. Even on VAR-d30, where improvements were marginal, we remark that IGG still yielded a marked increase in IS from 315.64 to 321.28 (without sacrificing FID), thereby attaining a more optimal quality-diversity trade-off than CFG.
>
> ### **2. IGG incurs negligible overhead compared to CFG**
>
> We first clarify that **IGG does not introduce any auxiliary network**. In fact, the only difference between IGG and CFG lies in its inference-time mechanism, where IGG applies an adaptive (instead of constant) guidance weight. On the other hand, the training process is identical for both methods.
>
> Regarding the concern about computational overhead, Table R1 details the inference time and GFLOP counts of CFG vs. IGG. The results indicate that, compared to CFG, **IGG incurs negligible computational overhead**. Indeed, this is expected since IGG solely introduces an additional dot-product attention operation at each scale.
>
> | Model         | Inference time (per image)  | GFLOPs (per image) |
> | ------------- | --------------------------- | ------------------ |
> | VAR-d30-*CFG* | 0.1391                      | 259.93             |
> | VAR-d30-*IGG* | 0.1408 (+0.0017)            | 259.93 (+0.00)     |
> | VAR-d36-*CFG* | 0.7009                      | 311.12             |
> | VAR-d36-*IGG* | 0.7103 (+0.0094)            | 311.12 (+0.00)     |
>
> *Table R1. Inference time and GFLOP counts. Compared to CFG, IGG incurs a mere 1% increase in inference time and no additional FLOPs.*
>
> ### **3. IGG is effective on non-SwAR models**
>
> We note that IGG is originally designed to tackle the misalignment of guidance that gradually arises as the model progresses from one scale to the next, a problem most relevant for SwAR models.
>
> Nevertheless, **IGG can indeed be extended to general AR models**. To verify this claim, we implemented our method on LlamaGen, a traditional token-wise model. To adapt to its next-token prediction approach, we modified the original formulation (see Section 5 in the manuscript) to causal attention, that is, every token attends only to the past tokens up to a set window size $s$. In our experiment, we set $s=1$ for simplicity.
>
> As reported in Table R5, compared to CFG, **IGG was able to attain a 2% improvement in FID**. Notably, it did so without worsening IS (even slightly improving it), indicating a more optimal quality-diversity trade-off.
>
> | Model                     | FID      | IS         | Prec     | Rec      |
> | ------------------------- | -------- | ---------- | -------- | -------- |
> | LlamaGen-L-*CFG* (w=2.00) | 3.07     | 256.06     | **0.83** | 0.52     |
> | LlamaGen-L-*IGG* (w=2.00) | **3.01** | **256.42** | **0.83** | **0.53** |
>
> *Table R5. Comparison of IGG against CFG on class-conditioned generation for ImageNet256 with LlamaGen-L as the backbone.*

---

> ### Author Response · Authors · 2025-11-26
> **Response to Reviewer ksrD [2/2]**
>
> ### **4. Clarification on the core method of IGG**
>
> We acknowledge that the realisation of IGG as a self-attention operation deserves elaboration, and we endeavour to do this in the upcoming revision of the manuscript. Nevertheless, we believe that the idea proposed in Section 5 is clear and well-motivated as is.
>
> Specifically, our analysis in Section 4 revealed that the naive adoption of CFG in SwAR models tends to cause **misaligned guidance as the model progresses from one scale to the next**. To eliminate this critical problem, we posit that an improved guidance method needs to automatically **identify and amplify guidance signals on semantically important tokens**. This led to the general formulation of IGG (Equation 5), $\tilde p_\theta(s_k|c) = p_\theta(s_k) + \gamma_k \cdot f_k(s_k|c) \cdot p_\theta^\to(s_k|c)$, where
>
> - $p_\theta^\to(s_k|c) := p_\theta(s_k|c) - p_\theta(s_k)$ are the guidance signals, and
> - $f_k$ is a function that adaptively determines the weight to apply to each token in $s_k$.
>
> To realise $f_k$, we remarked that semantically similar tokens tend to carry the same values. Thus, $f_k$ can be designed so as to align the value of a token to the values of tokens similar to it. This led to the proposal of the **self-attention operation on the guidance signals** (Equation 6). Indeed, it is widely known that attention computes, for each token, the average of all tokens weighted by their (dot-product) similarity to the subjected token.
>
> Regarding the reviewer's questions:
>
> ### **Q1. What is the computational cost of IGG per sampling step?**
>
> We believe this question has been addressed in Section 2 above.
>
> ### **Q2. Can IGG be applied to naive autoregressive models, such as LlamaGen?**
>
> We believe this question has been addressed in Section 3 above.
>
> We hope these answers have sufficiently addressed the reviewer's concerns and questions. Please do not hesitate to follow up for any further enquiries.

---

### Official Review · Reviewer_6KR7 · 2025-11-01

**Soundness:** 3
**Presentation:** 3
**Contribution:** 2
**Rating:** 4
**Confidence:** 4

**Summary:**

The idea central to this paper is that not all tokens require the same amount of guidance. The authors propose a novel, adaptive approach to improve the efficiency and quality of content generation by dynamically allocating the guidance strength based on the specific needs of each token during the generation process

**Strengths:**

1. The authors argue that Adaptive guidance is helpful which makes sense in some way

2. The results seems  to outperform cfg based sampling

3. Reinforcing informative patches in generation makes some sense in a way

**Weaknesses:**

1. IGG seems to introduce more complex form of hyperparameter

2. A table of comparison for comparative computation could be helpful

3.  There are papers with much better results for example https://arxiv.org/abs/2410.19324, so I do not think the claim the present method is SOTA is well supported

4. The comparison is very vague and does not say why the proposed method outperforms and what exactly the previous methods lack. Same goes for relatred work

5. With this new method what are the new potential biases that can affect, a discission on that will be meaningful

7. The method should evaluate on standard benchmarks like Compbench and Geneval to show the usefulness of this method

8. Side by side sensitivity analysis and statistical analyis of variance in results must be conducted for both cfg and igg

**Questions:**

See Weaknesses

---

> ### Author Response · Authors · 2025-11-26
> **Response to Reviewer 6KR7 [1/2]**
>
> We sincerely thank Reviewer 6KR7 for the constructive feedback. Regarding their concerns:
>
> ### **1. IGG uses the same hyperparameter as CFG**
>
> We clarify that **IGG admits the same hyperparameter as that of CFG**, namely a guidance weight $w$ that controls the "nudges" exerted on unconditioned tokens (more precisely, their corresponding logits). The difference lies in the nudges themselves: while CFG applies a constant nudge to every token, IGG automatically adapts its nudge based on the semantic importance of the subjected token.
>
> ### **2. IGG does not incur much more computation than CFG**
>
> To verify the titled statement, we report the inference time and GFLOP counts of CFG vs. IGG in Table R1. The results indicate that **the overhead incurred by IGG is minimal**. Indeed, this is expected since our method solely introduces an additional dot-product attention operation at each scale.
>
> | Model         | Inference time (per image)  | GFLOPs (per image) |
> | ------------- | --------------------------- | ------------------ |
> | VAR-d30-*CFG* | 0.1391                      | 259.93             |
> | VAR-d30-*IGG* | 0.1408 (+0.0017)            | 259.93 (+0.00)     |
> | VAR-d36-*CFG* | 0.7009                      | 311.12             |
> | VAR-d36-*IGG* | 0.7103 (+0.0094)            | 311.12 (+0.00)     |
>
> *Table R1. Inference time and GFLOP counts. Compared to CFG, IGG incurs a mere 1% increase in inference time and no additional FLOPs.*
>
> ### **3. IGG improves SOTA for AR models**
>
> The aim of this work is to improve the guidance process of AR generative models. Thus, **our SOTA claim pertains exclusively to AR models**, namely VAR, VAR-CLIP, LlamaGen, HART, and Switti (as reported in Tables 2 and 3). Indeed, as alluded to in the caption of Table 1, we acknowledge that there remains a performance gap between AR and diffusion models.
>
> ### **4. Why IGG outperforms previous methods**
>
> ### 4.1. What previous method lacks
>
> As highlighted in the last paragraph of Section 4, the naive adoption of CFG in SwAR models tends to cause misaligned guidance amongst tokens. We hypothesised that this stems from the fact that CFG applies a constant guidance weight across all tokens, which leads to the **dispersion of guidance signals as the model progresses from one scale to the next**. Figure 1 (top row), Figure 2 (bottom row), and Figure 6 (top rows) provide illustrative examples of this behaviour.
>
> ### 4.2. How IGG improves upon previous methods
>
> IGG outperforms CFG by addressing the shortcomings of guidance misalignment. In particular, it instead **applies an adaptive weight for each token based on its semantic importance**. This mechanism ensures that guidance is prioritised for regions where insufficient guidance can detrimentally affect the quality of the generated image.
>
> Regarding the section discussing related works (Section 2), our intention was to contextualise the adoption of diffusion-based CFG for SwAR models and, consequently, set up our analysis in Section 4.
>
> ### **5. IGG does not introduce new biases**
>
> Unfortunately, the reviewer did not make clear the precise type of biases they are referring to. Nevertheless, we emphasise that IGG differs from CFG exclusively in its inference-time mechanism (by applying an adaptive guidance weight, see above). On the other hand, the training process is identical to that of CFG. Thus, **IGG does not introduce any new bias** that may emerge from the training data.

---

> ### Author Response · Authors · 2025-11-26
> **Response to Reviewer 6KR7 [2/2]**
>
> ### **6. IGG performs well on GenEval**
>
> We remark that our method was indeed evaluated on the GenEval benchmark; see Table 3. Since the Overall score already provides a holistic reflection of model performance, we omitted the remaining scores in the manuscript for the sake of brevity. Should the reviewer want to see the full results, they are reported in Table R2.
>
> | Model          | Overall  | Single object | Two object | Counting  | Colors | Position  | Color attribution |
> | -------------- | -------- | ------------- | ---------- | --------- | ------ | --------- | ----------------- |
> | VAR-CLIP-*CFG* | 0.22     | 0.694         | 0.068      | **0.113** | 0.431    | 0.003     | 0.003             |
> | VAR-CLIP-*IGG* | **0.23** | **0.716**     | **0.078**  | 0.094     | **0.452**  | **0.008** | **0.005**         |
> | Switti-*CFG*   | 0.62     | 0.997         | 0.753      | 0.556     | 0.883    | 0.113     | 0.398             |
> | Switti-*IGG*   | **0.64** | **1.000**    | **0.808**  | **0.609** | **0.888**  | **0.123** | **0.430**         |
>
> *Table R2. Full evaluation of IGG on GenEval. VAR-CLIP generates 256x256 images while Switti generates 512x512 images.*
>
> Notably, **IGG consistently outperforms CFG in almost every metric**, with the only exception of the Counting score for VAR-CLIP. In particular, the biggest improvement was found in the "Two object" score, where a 15% and 7% increase was attained for VAR-CLIP and Switti, respectively. This result **highlights IGG's ability to ground important semantic information even in complex settings** where there are multiple foreground objects.
>
> ### **7. IGG exhibits similar variability to CFG**
>
> To compare the sensitivity of IGG against CFG, we ran the class-conditioned generation experiment (Table 2) of VAR-d30 across 10 different seeds. Table R3 reports the standard errors for each metric.
>
> | Method | FID   | IS    | Prec  | Rec   |
> | ------ | ----- | ----- | ----- | ----- |
> | CFG    | 0.005 | 0.538 | 0.000 | 0.002 |
> | IGG    | 0.009 | 0.627 | 0.000 | 0.002 |
>
> *Table R3. Standard errors of CFG and IGG in class-conditioned generation using VAR-d30.*
>
> These results indicate that **CFG and IGG exhibit similar variability across all metrics**. To verify this hypothesis, an additional F-test was conducted. The results are reported in Table R4. Since all p-values fall above 0.05, we concluded that the differences in variance between the two methods are not statistically significant.
>
> | Metric | F-statistic | p-value | p<0.05 |
> | ------ | ----------- | ------- | ------ |
> | FID    | 3.190       | 0.099   | False  |
> | IS     | 1.358       | 0.656   | False  |
> | Prec   | n/a         | 1.000   | False  |
> | Rec    | 2.095       | 0.286   | False  |
>
> *Table R4. Comparison of variances of CFG and IGG via an F-test. p<0.05 indicates statistical significance.*
>
> We hope these answers have sufficiently addressed the reviewer's concerns. Please do not hesitate to follow up for any further enquiries.

---

### Meta-Review · Area_Chair_7sJ2 · 2025-12-30

**Summary:**

The paper investigates the behavior of Classifier-Free Guidance (CFG) in Scale-wise Autoregressive (SwAR) models, identifying a "misalignment" issue where guidance signals become dispersed compared to diffusion models. To address this, the authors propose Information-Grounding Guidance (IGG), a training-free inference method that applies a self-attention mechanism to the guidance logits to concentrate signals on semantically salient regions.

**Reviewer Concerns:**

The reviewers initially raised concerns regarding:

1. Marginal Improvements: Reviewers ksrD and sSUr noted that the quantitative gains on standard benchmarks are modest (e.g., FID improvement from 1.93 to 1.92 on VAR-d30), questioning the practical significance of the method.
2. Methodological Simplicity: Reviewers found the method description "crude" and the contribution effectively limited to a heuristic attention map, questioning the depth of the innovation.
3. Computational Scalability: Reviewers sSUr and ksrD raised concerns about the cost of global self-attention at higher resolutions. While the authors proposed a sliding-window variant, it was not the primary method evaluated.
4. Inconsistent Validation: Reviewer 7N6s noted that the hypothesis (IGG works better at higher scales) was not consistently supported across all text-to-image benchmarks (e.g., Switti results).

Rebuttal and Remaining Concerns:
While the authors provided data showing minimal latency increase at current resolutions and extended experiments to LlamaGen, critical concerns remain unresolved:

1. Incremental Contribution: The consensus leans towards the view that the method is a minor heuristic modification rather than a fundamental contribution. The performance gains, while present, are often marginal and do not convincingly justify the introduction of a new inference hyperparameter/mechanism.
2. Scalability Bottleneck: Although the authors showed negligible overhead at , the AC agrees with the reviewers that the  complexity of global self-attention poses a significant theoretical bottleneck for scaling to higher resolutions, which contradicts the efficiency goals of AR models. The proposed "sliding window" fix was not rigorously established in the main experiments.
3. Lack of Strong Consensus: The rebuttal failed to significantly improve the scores of the critical reviewers (Reviewer ksrD maintained a 2, Reviewer 6KR7 maintained a 4).

**Reviewer Scores:**

The scores reflect a lack of enthusiasm for the paper's contribution.

1. Reviewer ksrD: 2 -> 2. Remained unconvinced by the magnitude of improvement and the "universality" of the method.
2. Reviewer 6KR7: 4 -> 4. Questioned the SOTA claims and hyperparameter complexity; did not improve score after rebuttal.
3. Reviewer sSUr: 6 -> 6. Acknowledged the clear diagnosis but noted the gains were "modest."
4. Reviewer 7N6s: 6 -> 6. Appreciated the insight but questioned performance inconsistencies.

---

### Decision · Program_Chairs · 2026-01-26

Reject